

# BiMGCL: rumor detection *via* bi-directional multi-level graph contrastive learning

Weiwei Feng[1], Yafang Li[2], Bo Li[1], Zhibin Jia[1] and Zhihua Chu[2]

[1] School of Computer Science and Engineering, Beihang University, Beijing, Beijing, China
[2] Faculty of Information Technology, Beijing University of Technology, Beijing, Beijing, China

## ABSTRACT

The rapid development of large language models has significantly reduced the cost of producing rumors, which brings a tremendous challenge to the authenticity of content on social media. Therefore, it has become crucially important to identify and detect rumors. Existing deep learning methods usually require a large amount of labeled data, which leads to poor robustness in dealing with different types of rumor events. In addition, they neglect to fully utilize the structural information of rumors, resulting in a need to improve their identification and detection performance. In this article, we propose a new rumor detection framework based on bi-directional multi-level graph contrastive learning, BiMGCL, which models each rumor propagation structure as bi-directional graphs and performs self-supervised contrastive learning based on node-level and graph-level instances. In particular, BiMGCL models the structure of each rumor event with fine-grained bidirectional graphs that effectively consider the bi-directional structural characteristics of rumor propagation and dispersion. Moreover, BiMGCL designs three types of interpretable bi-directional graph data augmentation strategies and adopts both node-level and graph-level contrastive learning to capture the propagation characteristics of rumor events. Experimental results on real datasets demonstrate that our proposed BiMGCL achieves superior detection performance compared against the state-of-the-art rumor detection methods.

## INTRODUCTION

The increasing use of social media has accelerated the distribution of misleading or unreliable information on a broad scale, which seriously jeopardizes the stability of the online environment (*Fu & Sui, 2022*). False information, particularly malevolent rumors, can mislead the public, have an impact on individual lives, undermine social order, and even have broad repercussions on the economy and politics at the federal level. Notably, the advent of OpenAI's ChatGPT has significantly lowered the cost of generating rumors, with News Guard considering it the most powerful tool for spreading false information in internet history (*Jack Brewster, 2023*). Consequently, the development of an effective rumor detection method is of utmost importance.

Corresponding author
Yafang Li, yafangli2014@163.com

Rumor detection has long been an important research area in natural language processing. Recently, there are various deep learning-based approaches proposed to combat the ever-changing landscape of rumor information. These methods can be broadly classified into feature-based detection methods and graph-based detection methods. Feature-based detection methods, including recurrent neural networks (RNNs) such as long short term memory (LSTM) and gated recurrent unit (GRU) networks, and their variants, effectively capture the temporal relationships between posts in rumor propagation chains (*Ma et al., 2016*; *Wu et al., 2020*). Convolutional neural network (CNN)-based methods excel at learning local spatial feature representations (*Rani, Das & Bhardwaj, 2021*; *Yu et al., 2017*), focusing on detecting rumors by leveraging text content, user features, and propagation patterns extracted from large datasets. However, these methods study isolated features and overlook the structural information of rumor propagation, making them susceptible to evasion by polymorphic rumor variants.

Inspired by recent advances in graph neural networks, methods like Bi-Directional Graph Convolutional Network (BiGCN) (*Bian et al., 2020*), Edge-enhanced Bayesian Graph Convolutional Network (EBGCN) (*Wei et al., 2021*), propagation path aggregating (PPA) neural network (*Zhang et al., 2021*), Graph-aware Co-Attention Networks (GCAN) (*Lu & Li, 2020*), Factual News Graph (FANG) (*Nguyen et al., 2020*), global-local attention network (GLAN) (*Yuan et al., 2019*), incorporate propagation structure information into rumor detection models. These graph-based methods learn more comprehensive representations and fully exploit the interaction between pieces of rumor information, revealing the inherent patterns of rumor propagation.

However, existing deep learning-based detection models rely heavily on large amounts of labeled data for supervision (*Liu et al., 2021*). To our best knowledge, acquiring high-quality training data with fine-grained labels, especially for rumor information, is challenging and time-consuming. Consequently, current deep learning-based detection models struggle to effectively detect advanced rumor variants.

Rumor propagation has evolved from known rumors to emergent rumors that arise alongside trending events on social media platforms. These rumor events tend to be sudden and may emerge from various fields with distinct backgrounds. Neural networks trained *via* supervised learning strongly depend on the data distribution of existing datasets, which makes them less robust when dealing with different rumor events. Moreover, the bi-directional structure of rumor events is often overlooked. Some existing methods, such as Rumor Detection on social media with Event Augmentations (RDEA) (*He et al., 2021*), rumor detection based on graph attention network (RDGAT) (*Lv et al., 2022*), Temporal Incorporating Structure Networks (TISN) (*Luo et al., 2022*), User-aspect Multi-view Learning with Attention for Rumor Detection (UMLARD) (*Chen et al., 2022*), ignore the simultaneous propagation and dispersion structures in rumor events. Learning solely from the propagation structure of rumors makes it difficult to obtain comprehensive and complete high-level structural information. Bi-directional rumor representations can extract structural information from multiple perspectives and augment the model's ability to discriminate rumor events. Lastly, current methods struggle to cope with label-sparse scenarios. In such situations, it is crucial to maximize the quality of model training with

limited dataset sizes. This raises the question of how to best utilize limited data and leads to the innovations presented in our work.

In this article, we propose BiMGCL, a new multi-level bi-directional graph contrastive learning approach for more robust and effective rumor detection. By processing both top-down and bottom-up rumor structure information, BiMGCL leverages bi-directional features of rumor propagation and dispersion. Additionally, self-supervised data augmentation methods are used for model pre-training. Ultimately, the trained model is used to detect and classify social media rumor events. Specifically, the bi-directional representation of the rumor event is obtained by composing graphs of the propagation and dispersion directions of the rumor event, respectively. We then use three effective data augmentation methods: Responded posts attribute masking, subsampling, and edge modification, which results in a rich set of sample instances. Next, we construct positive and negative sample pairs, where original samples and their augmented samples form positive pairs and different samples form negative pairs. Using a multi-level contrastive learning approach combined with the bi-directional graphs of rumor events, we pull the positive sample pairs closer while pushing negative sample pairs apart, ultimately completing the model pre-training. Additionally, during the convolution process, we concatenate the original root features in the feature matrix to augment the influence of the rumor root. Finally, we fine-tune the model using a portion of the rumor data to achieve a more robust rumor detection method with better generalization performance.

In summary, the contributions of this article are as follows:

- We propose a novel multi-level bi-directional graph contrastive learning framework, BiMGCL. To the best of our knowledge, this is the first study to introduce multi-level bi-directional graph contrastive learning into rumor detection tasks. The framework achieves efficient rumor detection by self-supervisedly identifying advanced rumor variants through comparing distinguishable patterns among instances of rumor propagation graphs.
- We design three types of bi-directional graph data augmentation methods, which take advantage of the causal features of top-down propagation of rumors through relational chains and the structural features of rumor dispersion within communities obtained through bottom-up aggregation, for generating complex high-level positive and negative bi-directional graph pairs, alleviating the inductive bias and improving the effectiveness of rumor detection.
- We conduct extensive experiments using the Twitter15 and Twitter16 datasets, demonstrating that BiMGCL outperforms other state-of-the-art (SOTA) methods.

## RELATED WORK

### Feature-based rumor detection

In recent years, because of the effectiveness of machine learning in handling huge numbers of labeled samples, feature-based approaches to rumor detection have been widely studied with the aim of extracting representative features to detect rumor events.

The most crucial step in traditional rumor detection methods is selecting and extracting prominent features from the data. *Castillo, Mendoza & Poblete (2011)* studied Twitter posts on popular topics and extracted 68 features related to user posting and retweeting, original post topics, and post text. They then used decision trees for automatic classification. *Yang et al. (2012)* were the first to analyze and detect rumors on Chinese microblog. They extracted two new features, posting location and posting client, from the data and combined them with information text and propagation structure features, using support vector machine (SVM) for classification. *Kwon et al. (2013)* investigated the time, structure, and text of rumor propagation and proposed a new cyclical time-series model. *Ma et al. (2015)* replaced manual feature extraction with a tree kernel method and proposed propagation tree kernel, a method extracts higher-order features that distinguish different types of rumors by comparing the similarities between propagation tree structures and then uses SVM for rumor recognition.

In recent years, many rumor detection methods have employed deep learning techniques for feature extraction and classification of rumors. *Ma et al. (2016)* were the first to use deep learning models for rumor detection on microblogs. They utilized the loops formed by connections between units in recurrent neural networks (RNNs) to capture dynamic temporal signal features of rumor propagation. Early rumor detection could be achieved through complex repeating units and additional hidden layers. Similar approaches include long short-term memory (LSTM) (*Liu & Wu, 2018*), gated recurrent units (GRU) (*Lu & Li, 2020*) and CNN (*Yu et al., 2017*). Because the temporal structure features can only represent the sequential propagation of rumors, these methods still have significant shortcomings. A large number of abundant and important structural features are also hidden by the spread of rumors, or the forwarding relationships.

However, most of the above-mentioned feature-based techniques overuse isolated features and are unable to identify the interaction between different rumor messages as well as the propagation patterns of rumors. To solve these problems, BiMGCL introduces bi-directional graph to simulate the interactions between different rumors, which can effectively capture the intrinsic interactions between rumor messages.

## Graph-based rumor detection

In contrast to feature-based methods, various graph-based approaches have been put forth recently. These approaches perform better because they place a greater emphasis on the structure data of rumor events. In order to capture rich structural information for rumor detection, the global-local attention network (GLAN) (*Yuan et al., 2019*) models the global relationship between all source tweets, retweets, and users as a heterogeneous graph. This method produces a better integrated representation for each source tweet by co-encoding local semantic and global structural information and fusing the semantic information of relevant retweets with attention mechanisms. The Factual News Graph (FANG) (*Nguyen et al., 2020*) enhances the quality of the representation by capturing more social structure and engagement patterns based on the heterogeneous network. To learn the correlation between the source tweet and retweet propagation as well as the co-influence between the source tweet and user engagement, A graph-aware co-attention network (GCAN) (*Lu &*

*Li, 2020*) develops a dual co-attention technique based on the graph structure. Rumor2vec (*Tu et al., 2021*) proposes the concept of the union graph to merge multiple propagation trees into a large graph when there are shared nodes and edges.

Graph convolutional networks (GCN) have been designed to enable neural networks to learn structured data representations. Based on CNNs, GCNs define convolution operations on graph structures, allowing nodes to integrate features with their neighborhoods. They are widely used in graph node classification and graph structure classification tasks. Bi-GCN (*Bian et al., 2020*) encodes and trains bi-directional rumor structures using graph convolutional networks, marking the first study to use GCNs for social media rumor detection. EBGCN (*Wei et al., 2021*) explore propagation uncertainty for rumor detection, which adaptively rethinks the reliability of latent relations by adopting a Bayesian approach. In recent studies, UMLARD (*Chen et al., 2022*) and TISN (*Luo et al., 2022*) employ GCN to learn the patterns of rumor propagation,they models rumor diffusion from multi-view perspective, using more information about rumors. However, they all relies on a large amount of labeled data, and supervised training struggles to cope with the real-time changes in social networks. RDEA (*He et al., 2021*) addresses this issue by using a self-supervised training approach, it expands data through various data augmentation methods and trains the encoding network using contrastive learning methods, ultimately achieving rumor event detection. However, similar to the issues present in the recent studies of UMLARD and TISN, they only utilizes the unidirectional propagation structure of rumor events, and thus, cannot deeply mine the dispersion structure information of rumor samples, resulting in significant limitations.

## METHODOLOGY

Motivated by the successes of contrastive learning in graph representation learning tasks (*Wang et al., 2020*; *Jin et al., 2021*; *Yu et al., 2022*; *Chen & Kou, 2023*), we propose the BiMGCL model as illustrated in Fig. 1 to study the effectiveness of the bi-directional multi-level graph contrastive learning for rumor detection. Concretely, our model mainly consists of four components: bi-directional graph construction, multi-strategy graph augmentation, multi-level contrastive learning-based rumor detection, and fine-tuning. Given the bi-directional graphs of the rumor samples from diverse rumor events, we attempt to map them to different clusters using a graphical isomorphic network (GIN) based contrastive learning method. Therefore, the key issues to be addressed by our proposed BiMGCL model are as follows.

**Issue 1**: In rumor detection tasks, how to model the fine-grained and comprehensive samples ('Bi-directional graph construction')? **Issue 2**: How should the effective positive and negative sample pairs for bi-directional graphs be constructed ('Multi-strategy graph augmentation')? **Issue 3**: How can the multi-level representations of the bi-directional rumor graphs be learned ('Multi-level contrastive learning-based rumor detection')? **Issue 4**: What are the discrimination rules ('Discriminator-based rumor detection')?

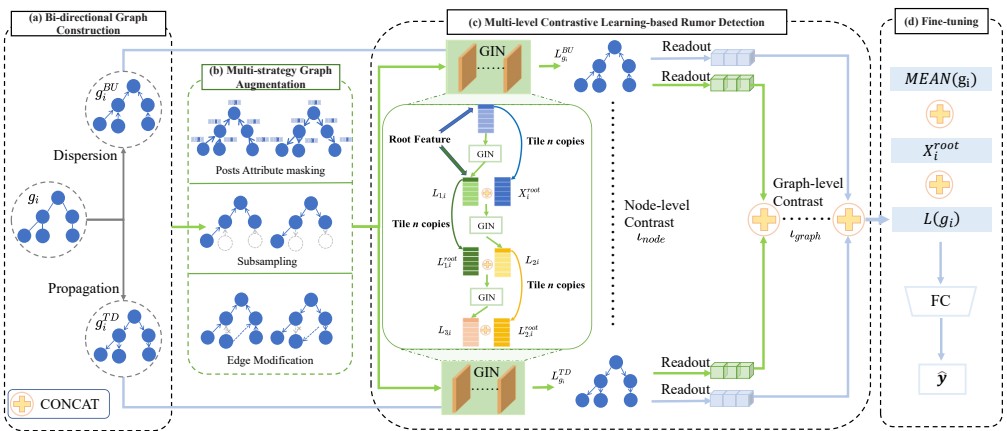

**Figure 1 The general framework of BiMGCL.** The framework is made up of the following four parts: (A) Bi-directional graph construction intends to model the propagation and dispersion of rumors using a bi-directional graph $g_i$ to construct a rumor propagation structure with rumor propagation and diffusion properties. (B) In multi-strategy graph augmentations, three forms of transformations, such as attribute masking, subsampling, and edge modification, are used to generate positive and negative instance pairs. (C) Multi-level contrastive learning-based rumor detection aims to evaluate the agreement between the nonlinear transformed representations of instance pairs by learning graph-level $\iota_{graph}$ and node-level $\iota_{node}$ comparisons *via* a contrastive loss. (D) Fine-tuning aims to fine-tune the pre-trained model for contrast learning and predicting the target rumor event.

## Problem definition

Assume that $C = \{c_1, c_2, \ldots, c_m\}$ is the dataset for rumor detection, where $c_i$ represents the $i$th rumor event. $c_i = \left\{ r_i, \omega_1^i, \omega_2^i, \ldots, \omega_{n_i-1}^i, G_i \right\}$, where $n_i$ denotes the number of posts in this event, $r_i$ is the source post, $\omega_j^i$ is the $j$th responsive post, and $G_i$ represents the propagation structure. Specifically, we use $G_i$ to denote a graph $\langle V_i, E_i \rangle$, where $V_i = \left\{ r_i, \omega_1^i, \omega_2^i, \ldots, \omega_{n_i-1}^i \right\}$ and $E_i = \left\{ e_{st}^i | s, \right\} \mathrm{t} = 0, \{\ldots, n_i - 1\}$ denotes the set of edges between responded posts, retweeted posts, or responsive posts, and $r_i$ is the root node (*Ma, Gao & Wong, 2018*). The feature matrix is denoted as $X_i = \left[ x_0^i, x_1^i, \ldots, x_n^i \right]$, where $x_j^i \in \mathbb{R}^d$ represents the $j$th element of $X_i$ and $d$ denotes the dimension of the feature. Let $A_i \in \{0, 1\}^{n_i \times n_i}$ be the adjacency matrix with the initial value as

$$a = \begin{cases} 1, & if \ e_{st}^i \in E_i \\ 0, & otherwise \end{cases}. \tag{1}$$

Additionally, our goal is to train a classifier $f : C \to Y$, where $C$ represents the set of events and $Y$ denotes the fine-grained classes $\{N, F, T, U\}$ (*i.e.,* Non-rumor, False Rumor, True Rumor, and Unverified Rumor). Furthermore, during the model training phase, we generate $\hat{G}$ through data augmentation, and subsequently a classifier $f(\cdot)$ is trained together with the original $G$.

## Bi-directional graph construction

This article particularly focuses on global features based on the overall structure of rumour propagation direction, which is an important cue for rumour detection. The depth

propagation features and breadth propagation features of rumours play an important role in rumour detection. Unfortunately, traditional temporal structure features and single rumour spread features ignore the overall structural features of rumour events, and the inability to exploit these meaningful interactions of information from different propagation directions leads to a degradation in the performance of these methods. Obviously, bi-directional graphs are a powerful tool for modelling complex interactions between information and relationships in different directions. Therefore, to settle Issue1, using bipartite graphs to model the deep spread of rumours in relational chains and the wide spread of rumours in social groups is an elegant choice for the rumour detection task.

We construct two different directions for each sample $g_i$: a top-down direction of propagation and a bottom-up direction of dispersion, *i.e.,* for the direction of rumor propagation, if $\omega_2^i$ has a response to $\omega_1^i$, there will be an directed edge $\omega_1^i \rightarrow \omega_2^i$, while for the direction of dispersion, there will be an directed edge $\omega_2^i \rightarrow \omega_1^i$. We use $g_i^{TD}$ and $g_i^{BU}$ to represent these two cases, where the adjacency matrix $A_i^{TD}$ in $g_i^{TD}$ contains the top-down part of $A_i$ and the adjacency matrix $A_i^{BU}$ in $g_i^{BU}$ contains the bottom-up part of $A_i$. Also the same feature matrix $X_i$ is used for both directions, *i.e.,* $X_i^{TD} = X_i^{BU} = X_i$.

Finally, as shown in Fig. 1A, we get the fine-grained bi-directional graphs of rumor samples, which are represented as adjacency matrices $A_i^{TD}$, $A_i^{BU}$, and attribute matrix $X_i$.

## Multi-strategy graph augmentation

Recent research has shown that the performance of contrast learning is directly influenced by the samples of positive and negative instance pairs (*Yin et al., 2022*; *Miao et al., 2022*; *Hassani & Khasahmadi, 2020*). BiMGCL differs from existing contrastive learning methods (*Li et al., 2022*; *Zhu et al., 2021*; *Qiu et al., 2020*; *You et al., 2020*) which only deals with geometric information in graphs or pure dependencies in a single isomorphic graph, it aims to produce realistic rumor propagation augmented instances using both top-down and bottom-up types. Therefore, to resolve issue 2, we take into account both the bi-directional graph's structure and attribute information to generate augmented positive and negative instances. By imposing structural prior or semantic prior, we focused on two directions of rumor propagation graph data augmentation methods. As illustrated in Fig. 1, responded posts attribute masking, which merely modifies the attribute matrix, is a semantic level augmentation method. On the contrary, subsampling and edge modification belong to structure-level augmentation, and they transform the adjacency matrix.

### *Responded posts attribute masking*

As shown in Fig. 1B, we extend the node attribute masking in adaptive graph augmentation (AGA) (*Wang et al., 2023*) and graph contrastive learning (GraphCL) (*You et al., 2020*) to transform the attribute matrix of rumor sample trees. Malicious users intentionally spread rumors during rumor propagation, usually motivated by financial or non-financial benefits (such as power and fame), while gullible users are vulnerable, everyday users who unintentionally contribute to the spread of false rumors because they mistake the false rumors for the truth. Therefore, placing too much faith on those who spread rumor information could have negative consequences. To solve this problem, we mask the

propagation tree's node features at random throughout each training stage, with the exception of the root node. For a given rumor instance graph $g_i$ in the Twitter datasets with its $X_i$ and $A_i$, denoted as $(A_i, X_i)$, we formulate the transformation function $\Gamma^{mask}$ as:

$$\Gamma^{mask}(A_i, X_i) = (A_i, \Gamma^{mask}(X_i)). \tag{2}$$

Formally, we use $\Gamma^{mask}$ to randomly mask a node attribute vector $X_i$:

$$\Gamma^{mask}(X_i) = X_i * (1 - M_i) + V * M_i \tag{3}$$

where $M_i \in \{0, 1\}$ indicates whether the node attribute is masked, generated with a certain probability. $V \sim N(\mu, \sigma^2)$ refers to the masking Gaussian noise, which is used to randomly mask the attributes of certain responded posts nodes for a given node feature matrix.

### Subsampling

Research has found that considering the entire lifecycle of a rumor is crucial for accurately understanding user behavior and responses. However, the early stages of a rumor's lifecycle may exhibit different characteristics. Users' initial reactions to rumors seem to lean more towards supporting the rumor, regardless of whether it is true or false. This highlights a potential challenge for models in detecting rumors at early stages. If the model has too much information about the entire lifecycle of the event, it may struggle to detect rumors early on, as users' behaviors and reactions change over time.

Inspired by the successes of subgraph sampling (*You et al., 2020*) or uniform sampling (*Xie et al., 2022*), we propose a custom random sampling approach for rumor trees that naturally picks up on the more distinct patterns of various responded posts.

Specifically, given a graph $g_i$, we start a random walk from its root node, moving to its neighbors with a certain probability $p_{move}$ and iterating multiple times, ultimately obtaining the sampled subgraph $g_{i\_sampling}$.

### Edge modification

Research suggests that when rumors reappear repeatedly, they may become increasingly real in people's perception, as a message that is widely circulated and repeatedly reinforced is regarded as reliable information. To address this issue, we employ edge modification (EM) (*Hu et al., 2020*; *You et al., 2020*; *Zhu, Du & Yan, 2020*; *Jin et al., 2021*; *Zhang et al., 2020*; *Zeng & Xie, 2021*), which changes the graph structure by randomly deleting and inserting a portion of edges, partially perturbing the given graph's adjacency, resulting in changes in the propagation path and node correlation of rumors in the network, thus effectively avoiding the recurrence of information.

Specifically, in the structure of rumor trees, in addition to deleting some edges in the adjacency matrix, we also add edges so that the number of edges deleted and added is equal. The advantage of doing this is that the graph can become more complex by adding extra edges while preserving the original feature attributes. For example, given a modification ratio $p$, randomly delete $p/2$ of the edges in the original adjacency matrix $A$ and randomly add the same proportion of new edges, making the number of edges deleted and added equal. The edge deletions and additions in this process are performed independently and

identically distributed. In particular, given a rumor graph $(A_i, X_i)$, the transformations function $\Gamma^{EM}$ as:

$$\Gamma^{EM}(A_i, X_i) = (\Gamma^{EM}(A_i), X_i) \qquad (4)$$

We define this process as follows, in which $\circ$ denotes Hadamard product:

$$\Gamma^{EM}(A_i) = M_{drop} \circ A_i + M_{insert} \circ (1 - A_i) \qquad (5)$$

By generating $M_{drop}$ and $M_{insert}$ according to whether the values in $A_i$ and $1 - A_i$ are equal to 1, randomly masking a portion of the elements with the same probability $p_{mod}$, we obtain the masking matrices representing discarded edges and added edges, respectively. Ultimately, some edges are deleted and added with the same probability.

Figure 1B shows three augmented methods. Given the original instance $g_i$ and the augmented instance $aug(g_i)$, the set of graphs obtained after augmentation is defined as $AG_i = g_i, aug(g_i)$. BiMGCL forms positive instance pairs by combining each original instance and its corresponding augmented instance, $(g_i, aug(g_i))^+$. In addition, randomly selected instances from other samples are used as negative instance pairs, forming $(AG_i, AG_j)^-$. Rumor trees adopt both top-down and bottom-up directions. Therefore, when performing data augmentation, both directions of rumor trees should be augmented simultaneously to ensure consistency between the data.

## Multi-level contrastive learning-based rumor detection

To address issue 3, we use a graph encoder to learn graph-level low-dimensional representations of different rumor propagation instances. We utilize the Graph Isomorphism Network (GIN) (*Xu et al., 2018*), a powerful graph neural network, as the graph encoder in BiMGCL. GIN has been proven to outperform other GNNs, such as GCN (*Kipf & Welling, 2016*) and GraphSAGE (*Hamilton, Ying & Leskovec, 2017*).

After the original rumor tree samples have undergone the three types of augmentations, a rich set of sample instances has been generated. We use the GIN network to learn the hidden structures and semantic information in these tree structures, with the detailed architecture shown in Fig. 1C. In this process, we place great emphasis on the root node's features, as the source posts of rumor events often contain rich information. By making full use of this, the neural network can learn more accurate abstract semantics of rumor propagation.

In Fig. 1C, for the $i$th rumor graph sample $g_i = (A_i, X_i)$, where $A_i$ includes both $A_i^{TD}$ and $A_i^{BU}$ (TD and BU represent the two directions of the rumor tree: Propagation (Top-Down) and Dispersion (Bottom-Up), as shown in Fig. 1A), we adopt the same encoder architecture for the bi-directional structure. First, we encode the $i$th sample as follows:

$$L_{1,i} = LeakyReLU(W_1^T(A_i\, X_i) + b_1) \qquad (6)$$

$$\tilde{L}_{1,i} = CONCAT(L_{1,i},\, X_i^{root}). \qquad (7)$$

In this process, $L_{1,i}$ represents the hidden layer features of the sample after the first layer of graph convolution, and $W_1^T$ and $b_1$ are parameters of the neural network. LeakyReLU

is used here as the activation function. The feature vector of each node after encoding is subsequently concatenated with the feature vector $R_i \in \mathbb{R}^{d \times 1}$ of the root node of the sample before encoding, which is used to augment the features of the original post of the rumor event. $X_i^{root} = [R_i, R_i, \ldots, R_i]^T$, $X_i^{root} \in \mathbb{R}^{n \times d}$ serves as the root node feature matrix to be concatenated for the $i$th sample, $d$ is the node feature dimension. Afterward, $\tilde{L}_{1,i}$ is convolved multiple times to extract sample information, similar to the aforementioned process:

$$L_{2,i} = LeakyReLU(W_2^T (A_i \tilde{L}_{1,i}) + b_2) \tag{8}$$

$$\tilde{L}_{2,i} = CONCAT(L_{2,i},\ L_{1,i}^{root}) \tag{9}$$

$$L_{3,i} = LeakyReLU(W_3^T (A_i \tilde{L}_{2,i}) + b_3) \tag{10}$$

$$L_{g_i} = \tilde{L}_{3,i} = CONCAT(L_{3,i},\ L_{2,i}^{root}). \tag{11}$$

After passing the $i$th sample $g_i$ through the same encoder $L$, we can obtain the bi-directional graph encoding representation of $g_i$ as $L_{g_i} = (L_{g_i}^{TD}, L_{g_i}^{BU})$. A multi-level contrast is then performed, taking into account both the property augmentation of the local structure and the global topological high-level information.

For the node-level contrastive loss, we consider each direction of the bi-directional graph separately, following the steps below. For sample $g_i$, we construct positive instance pairs using the encoded original graph $L_{g_i}$ and its augmented graph $L_{aug(g_i)}$ as: $(L_{g_i}, L_{aug(g_i)})^+$. Any distinct samples form negative instance pairs: $(L_{AG_i}, L_{AG_j})^-$. We use $\iota_{node}$ to denote the node-level contrastive loss. The contrastive loss for the positive instance pair of $g_i$ is defined as $\iota_{node}^{pos}(g_i)$, and the contrastive loss for the negative instance pair is defined as $\iota_{node}^{neg}(g_i)$. Here, $sim(\cdot, \cdot)$ represents the similarity between different sets of sample encodings, which is measured by constructing the Cartesian product of the two sets and using cosine similarity to measure the distance between each pair. The sum of the resulting values represents the similarity between the two sets.

$$\iota_{node}^{pos}(g_i) = -log\left(\frac{\exp(sim(L_{g_i}, L_{aug(g_i)})/\tau)}{\sum_{j=1}^{n} \exp(sim(L_{AG_i}, L_{AG_j})/\tau)}\right) \tag{12}$$

$$\iota_{node}^{neg}(g_i) = \frac{\sum_{i \neq j, j=1}^{n} \exp(sim(L_{g_i}, L_{AG_j})/\tau)}{\sum_{j=1}^{n} \exp(sim(L_{AG_i}, L_{AG_j})/\tau)}. \tag{13}$$

Inspired by infoNCE (*Oord, Li & Vinyals, 2018*), each contrast loss can be divided into upper and lower parts. The goal of $\iota_{node}^{pos}(g_i)$ is to increase the similarity between $L_{g_i}$ and $L_{aug(g_i)}$, while decreasing the similarity between different graphs $L_{AG_i}$ and $L_{AG_j}$. The $\tau$ is a temperature parameter that is often used to modulate the sharpness of the probability

distribution obtained from the similarity scores. Also, $\iota_{node}^{neg}(g_i)$ denotes a more exclusive encoding between $g_i$ and $AG_j$ in the context of decreasing the similarity between different graphs $L_{AG_i}$ and $L_{AG_j}$.

By summing the contrastive losses of all samples, we obtain the final node-level contrastive loss $\iota_{node}$. The smaller this loss, the closer the distance between the encodings of sample $g_i$ before and after augmentation, and the farther the distance between the encodings of $g_i$ and other distinct samples:

$$\iota_{node} = \frac{1}{2N}\sum_{i=1}^{n}(\iota_{node}^{pos}(g_i) + \iota_{node}^{neg}(g_i)). \tag{14}$$

For the graph-level contrastive loss, given the bi-directional graph encoding representation of $g_i$ as $L_{g_i} = (L_{g_i}^{TD}, L_{g_i}^{BU})$, we use the mean pooling operator to aggregate information from both sets of encoding representations:

$$S_{g_i} = CONCAT(MEAN(L_{g_i}^{TD}), MEAN(L_{g_i}^{BU})). \tag{15}$$

Similar to the node-level contrastive loss, to pool the augmented samples, we obtain $S_{aug(g_i)}$. The goal is to minimize the distance between $S_{g_i}$ and $S_{aug(g_i)}$, while maximizing the distance between different samples and their augmented graphs. We define the contrastive loss for the positive instance pairs as $\iota_{graph}^{pos}(g_i)$, and for the negative instance pairs as $\iota_{graph}^{neg}(g_i)$:

$$\iota_{graph}^{pos}(g_i) = -\log\left(\frac{\exp(sim(S_{g_i}, S_{aug(g_i)})/\tau)}{\sum_{j=1}^{n}\exp(sim(S_{AG_i}, S_{AG_j})/\tau)}\right) \tag{16}$$

$$\iota_{graph}^{neg}(g_i) = \frac{\sum_{i\neq j, j=1}^{n}\exp(sim(S_{g_i}, S_{AG_j})/\tau)}{\sum_{j=1}^{n}\exp(sim(S_{AG_i}, S_{AG_j})/\tau)} \tag{17}$$

we can compute $\iota_{graph}$:

$$\iota_{graph} = \frac{1}{2N}\sum_{i=1}^{n}(\iota_{graph}^{pos}(g_i) + \iota_{graph}^{neg}(g_i)). \tag{18}$$

Finally, by combining the node-level contrastive loss $\iota_{node}$ with the graph-level contrastive loss $\iota_{graph}$, our multi-level contrastive loss can be represented as:

$$\iota = \iota_{node} + \lambda\iota_{graph} \tag{19}$$

where $\lambda$ is a parameter to be adjusted in order to balance the importance of $\iota_{node}$ and $\iota_{graph}$.

## Discriminator-based rumor detection

After multiple rounds of pre-training to minimize the multi-level contrastive loss, we obtain a high-performance encoder. This encoder can be used to encode a rumor event, which is represented as:

$$L_{g_i} = CONCAT(L_{g_i}^{TD}, L_{g_i}^{BU}). \tag{20}$$

We employ a multi-layer perceptron (MLP) for rumor detection classification. In addition to using the encoding $L_{g_i}$ obtained from the model, we also utilize the root feature of the rumor event $X_i^{root}$ and the graph-level representation $MEAN(g_i)$ of the original rumor tree as additional information. Ultimately, we use these three parts as inputs and employ the MLP and softmax to perform classification.

$$I_i = CONCAT(L_{g_i}, X_i^{root}, MEAN(g_i)) \tag{21}$$

$$\hat{y}_i = softmax(MLP(I_i)). \tag{22}$$

Then, we use the cross-entropy loss function to measure the distance between the predictions $\hat{y}$ and the ground truth $y$ for all rumor events:

$$\mathcal{L}(y, \hat{y}) = -\sum_{i=1}^{N} y_i log(\hat{y}_i) + \lambda ||\Theta||_2^2. \tag{23}$$

In the loss function, $||\Theta||_2^2$ represents the L2 regularization term for the model parameters, and $\lambda$ is a balancing factor.

# EXPERIMENTS

## Experimental settings
### Datasets

To evaluate the spread of rumors, we utilized two publicly available Twitter datasets, Twitter15 and Twitter16 (*Ma, Gao & Wong, 2017*). The nodes in these datasets represent users, and the edges represent forwarding or response relationships. Features were extracted from the top 5,000 words based on their TF-IDF values. The statistics are presented in Table 1. Twitter15 and Twitter16 contain 1,490 and 818 claims, respectively, each labeled as Nonrumor (NR), False Rumor (F), True Rumor (T), or Unverified Rumor (U). '# of postings' is the number of all posts in the dataset, and '# of users' is the number of all users who participated in the posting. According to the veracity tag (*Ma, Gao & Wong, 2017*) of the associated item on rumor-dispelling websites like snopes.com and Emergent.info, the labels of each event in Twitter15 and Twitter16 were annotated. In other words, the label of these events was determined by the veracity tag of the article on the debunking websites.

## Experimental setup

We make comparisons with the following state-of-the-art baselines:

- Decision tree classification (DTC) (*Castillo, Mendoza & Poblete, 2011*): A decision tree model that combines various news features.
- Support Vector Machine based on Time Series (SVM-TS) (*Ma et al., 2015*): A linear SVM classifier that builds a time-series model using custom features.
- BU-RVNN (*Ma, Gao & Wong, 2018*): The structural recursive neural model based bottom-up tree neural network (with GRU unit) is used by the rumor detection method to learn about rumor information.

**Table 1  Statistics of the datasets.**

| Dataset | Twitter15 | Twitter16 |
|---|---|---|
| # of events | 1,490 | 818 |
| # of false rumors | 370 | 205 |
| # of true rumors | 372 | 205 |
| # of unverified rumors | 374 | 203 |
| # of non-rumors | 374 | 205 |
| # of postings | 331,612 | 204,820 |
| # of users | 276,663 | 173,487 |

- TD-RVNN (*Ma, Gao & Wong, 2018*): The structural recursive neural model based top-down tree neural network (with GRU unit) is used by the rumor detection method to learn about rumor information.
- Graph-aware Co-Attention Networks (*Lu & Li, 2020*): The rumor detection method proposes a GCN-based model that can characterize the rumor propagation mode and use the dual co-attention mechanism to capture the interaction between source text, user attributes, and propagation path.
- BiGCN (*Bian et al., 2020*): The rumor detection approach uses a GCN-based model to represent the overall structure of the rumor tree by utilizing its two primary characteristics, rumor propagation and dispersion.
- RDEA (*He et al., 2021*): A GNN model based on contrastive learning, which uses data augmentation to enable the model to do better at learning rumor propagation patterns.
- UMLARD (*Chen et al., 2022*): A multi-view learning rumor detection model proposed with a distinguishable feature fusion mechanism, acquiring user-aspect features, and incorporate them with content features.
- TISN (*Luo et al., 2022*): A rumor detection method considers time information and propagation structure, which Transformer encoder is used to extract text information, and GCN is used to learn the patterns of rumor propagation.

To provide a fair comparison, we randomly divided the datasets into five sections and performed 5-fold cross-validation to achieve robust results. We assess the accuracy (Acc.) over the two categories and the F1 measure (F1) for each class for the two Twitter datasets.

In the experiment, BiMGCL uses the Adam optimizer (*Kingma & Ba, 2014*). The feature length of each node in the feature matrix, drop rate in edge modification, and dropout rate are set to 64, 0.4, and 0.5, respectively. The epoch of self-supervised training is set to 25, while the supervised fne-tuning process is iterated upon 100 epochs, and early stopping (*Yao, Rosasco & Caponnetto, 2007*) is applied when the validation loss stops decreasing by 10 epochs. The software versions we use are Python-3.7, PyTorch-1.8.1 (CPU), pytorch_geometric-1.7.0. In the graph construction phase we use the APIs provided by pytorch_geometric. In the data augmentation phase, both the topology and node attributes of the graph are augmented to produce richer samples. The code for our implementation can be found in https://github.com/vivian-166/BiMGCL/ and we provide a link to raw data: https://github.com/vivian-166/BiMGCL/tree/main/data.

**Table 2  Experimental results on the Twitter15.**

| Method | ACC | F1 | | | |
|---|---|---|---|---|---|
| | | NR | FR | TR | UR |
| DTC | 0.454 | 0.733 | 0.355 | 0.317 | 0.415 |
| SVM-TS | 0.544 | 0.796 | 0.472 | 0.404 | 0.484 |
| BU-RVNN | 0.708 | 0.695 | 0.728 | 0.759 | 0.653 |
| TD-RVNN | 0.723 | 0.682 | 0.758 | 0.821 | 0.654 |
| GCAN | 0.808 | 0.866 | 0.759 | 0.812 | 0.690 |
| BiGCN | 0.836 | 0.791 | 0.842 | 0.887 | 0.802 |
| RDEA | 0.853 | 0.828 | 0.856 | 0.902 | 0.816 |
| UMLARD | 0.857 | 0.840 | 0.848 | 0.906 | 0.835 |
| TISN | 0.865 | 0.867 | 0.871 | 0.847 | 0.791 |
| **BiMGCL** | **0.882** | **0.868** | **0.885** | **0.918** | **0.850** |

**Notes.**
Bold indicates the best performance.

**Table 3  Experimental results on the Twitter16.**

| Method | ACC | F1 | | | |
|---|---|---|---|---|---|
| | | NR | FR | TR | UR |
| DTC | 0.465 | 0.643 | 0.393 | 0.419 | 0.403 |
| SVM-TS | 0.574 | 0.755 | 0.420 | 0.571 | 0.526 |
| BU-RVNN | 0.718 | 0.723 | 0.712 | 0.779 | 0.659 |
| TD-RVNN | 0.737 | 0.662 | 0.743 | 0.835 | 0.708 |
| GCAN | 0.765 | 0.799 | 0.754 | 0.678 | 0.784 |
| BiGCN | 0.864 | 0.788 | 0.859 | 0.932 | 0.864 |
| RDEA | 0.877 | 0.817 | 0.872 | 0.932 | 0.876 |
| UMLARD | 0.878 | 0.826 | 0.834 | 0.936 | 0.801 |
| TISN | 0.870 | 0.827 | 0.786 | 0.792 | 0.820 |
| **BiMGCL** | **0.890** | **0.829** | **0.884** | **0.942** | **0.894** |

**Notes.**
Bold indicates the best performance.

## Results and analysis

In this section, we evaluate the performance of the proposed framework for rumor detection by comparing it with the baseline methods. According to Tables 2 and 3, which display the experimental findings, our proposed BiMGCL beats all baseline approaches in terms of all assessment measures. In actuality, the following characteristics are what led to BiMGCL's improvement.

First, our proposed BiMGCL not only models various rumor post objects and their comments as graph structures to obtain information about the contextual structure of their propagation patterns, but also considers propagation and dispersion as two crucial characteristics of rumors, which facilitates the learning of more fine-grained propagation

patterns. This is in contrast to feature-based supervised learning rumor detection methods such as SVM-TS and DTC. In other words, these structural and semantic characteristics, which were discovered from the bi-directional graph of rumors, aid in enhancing the discriminability of rumor events, which results in the desired performance of BiMGCL. It is worth noting that our proposed BiMGCL improves 1.2%–42.8% over the feature-based supervised learning approach in terms of accuracy.

Second, to train the model in a supervised manner, most existing graph-based supervised learning rumor detection algorithms rely on a large corpus of labeled data (*Sun et al., 2022*; *Wei et al., 2023*). However, due to weak generalization for new target classes, such techniques fall short of detecting out-of-sample rumor events. In particular, GCAN, BiGCN, UMLARD and TISN employing graph neural networks to learn low-dimensional representations from rumor propagation trees, which typically pose a serious detection performance when the availability of training samples or labeling information is limited. Although UMLARD and TISN combined more information about rumors, they did not consider the relationship between different rumor events. Our proposed BiMGCL, in contrast to existing approaches, shifts the emphasis from labeling information to learning more sophisticated and discriminative patterns of rumor information propagation using multi-level contrastive learning. As a result, BiMGCL demonstrates improved rumor detection capabilities.

Third, BiMGCL has two advantages over the most relevant self-supervised work RDEA. On the one hand, RDEA investigates oversimplified data augmentation for node-to-node comparison learning, which applies to undirected graphs because they ignore nonlinear and hierarchical dependencies between rumor propagation patterns, leading to poor performance in out-sample rumor detection. In contrast, BiMGCL designs three insightful bi-directional graph data augmentation methods to identify more advanced rumor information, leveraging the rumor propagation and diffusion discriminant patterns, combining semantic and structural priors to increase the efficiency of rumor detection. Besides, compared to the node-to-node comparison learning approach in RDEA, we design a more expressive multi-level comparison approach to help graph encoders extract rumor knowledge from historical representations, which not only provide a stronger regularization to our bootstrapping objective but also enrich the self-supervision signals during the optimization.

Furthermore, the originating post of a rumor event often contains a wealth of detailed information and can wield substantial influence. However, UMLARD and TISN tend to overlook the crucial value embedded in the initial post's information during the propagation process. Conversely, BiMGCL stands out for its adeptness at harnessing the data present in the source post. This enables BiMGCL to thoroughly excavate the structural nuances within rumor trees, thereby significantly enhancing the efficacy of rumor detection.

### Ablation experiments

We further conducted ablation experiments to analyze the effect of each variant of BiMGCL, investigating the effect of each module in the pretraining encoder.

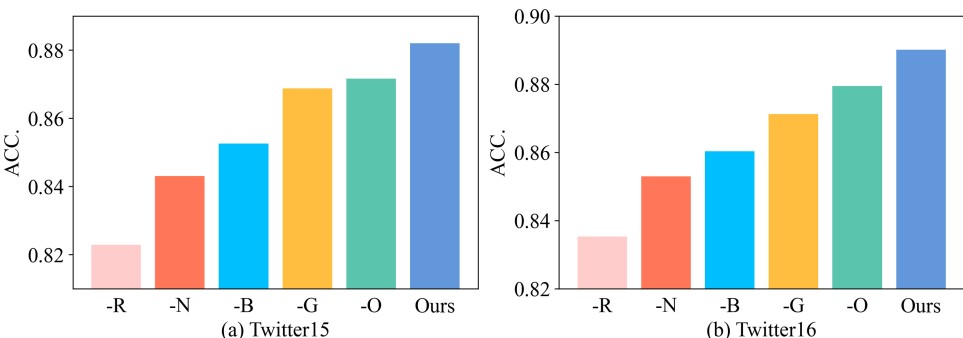

**Figure 2  The results of model method ablation experiments.**

The empirical results are summarized in Fig. 2. -O, -R, -B, -G, and -N represent our model without concatenating the original graph, without concatenating the root feature, without using bi-directional graph, without graph-level contrastive loss, and without node-level contrastive loss, respectively. We make the following three observations. First, the performance of BiMGCL decreases if any of these four components are missing, indicating that they are all crucial for BiMGCL. Second, the performance drop is the largest for -R, suggesting that root features are indispensable, and the source post plays an essential role in rumor detection. At the same time, the -B performance decreases significantly, which implies the importance of considering both top-down representation and bottom-up representation.

In addition to the ablation experiments on the methods used in the comparative learning model, we further conduct experiments on the data augmentation methods to explore their effectiveness. The results of the experiments are shown in Fig. 3, where 'NO AUG' represents that no data augmentation method is used, 'AM', 'SS', and 'EM' represent attribute masking, subsampling, and edge modification, respectively, which are the three data augmentation methods described in the above article, and 'ALL AUG' indicates that the three methods are used at the same time. It can be observed that when no data augmentation methods are used, the lack of training samples and insufficient data features have a greater impact on model training and fail to achieve better accuracy. This problem is partially alleviated when using these methods alone, and better results can be achieved by using all three augmentation methods together. This is because the data augmentation methods, on the one hand, expand the existing data and produce a large amount of data from the original graph, which enables the comparative learning discriminator to better distinguish the features between similar and different rumor graphs; on the other hand, these three data augmentation methods augment the data in terms of the graph features, graph localization information, and graph structure, respectively, and produce rumor graph samples that contain various semantics, which provides important supports for model training.

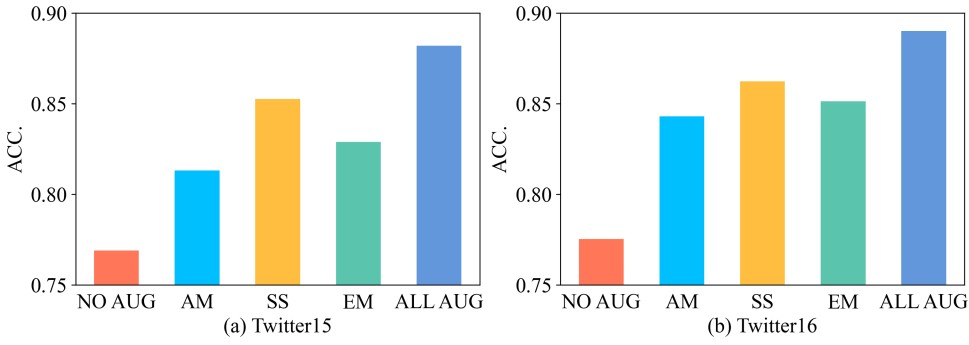

**Figure 3** The results of graph augmentation ablation experiments.

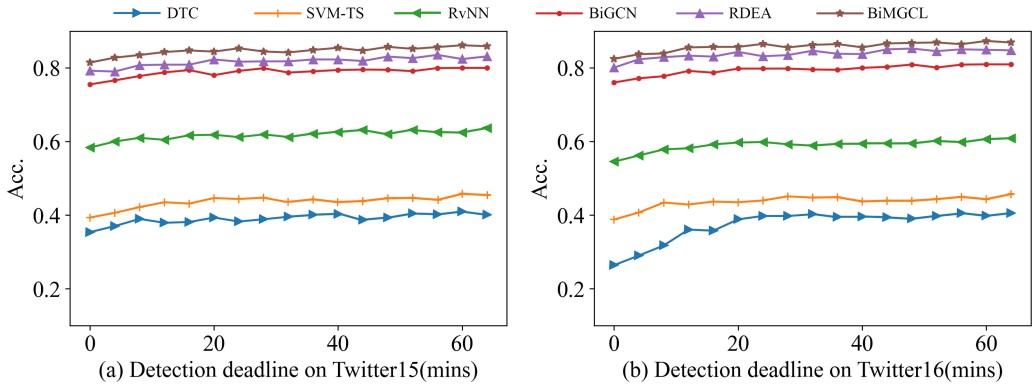

**Figure 4** Result of rumor early detection on Twitter15 (A) and Twitter16 (B).

## Early rumor detection

A crucial method of assessing the model is early rumor detection, which aims to identify rumors before they become widespread and have negative social effects. To build the early detection task, we set a series of detection deadlines and only use posts that were published before the deadlines to assess the accuracy of the proposed and baseline methods and confirm whether the model can accurately identify rumors based on the scant information carried by the current early moments.

The early rumor detection task is illustrated in Fig. 4 using the results of our BiMGCL method and several benchmarks. The BiMGCL method described in this research achieves a high level of accuracy very quickly after the source post-original broadcast. BiMGCL performs significantly better than the other models at each deadline, indicating that our method is not only beneficial for long-term rumor detection, but also more beneficial for early detection of rumors. Notably, our BiMGCL model outperforms other benchmark tests and is outstanding and consistent at all times, demonstrating the effectiveness of the combination of a bi-directional graph model and multi-level contrast learning for robust and accurate identification.

## CONCLUSION

In this article, we propose BiMGCL, a rumor detection framework based on bi-directional graph multi-level comparative learning, to achieve the goal of rumor detection for well-designed rumors. BiMGCL employs bi-directional graphs to simulate the propagation and diffusion patterns among rumor messages and captures distinct rumor properties through multi-level self-supervised comparative learning. We thoroughly assess BiMGCL's efficacy and efficiency, and the experimental findings validate that our proposed BiMGCL is superior at identifying rumors.

Although we have achieved encouraging results, with the development of multimedia technology, rumors in current social networks often contain information in multiple modalities, which calls for further research. In our future work, we plan to incorporate multimodal information into our graph neural network models, paying more attention to external factors influencing rumor propagation and the interpretability of model decisions.

### Funding

This work was supported by the National Natural Science Foundation of China under grant 62006009. The funders had no role in study design, data collection and analysis, decision to publish, or preparation of the manuscript.

### Grant Disclosures

The following grant information was disclosed by the authors:
The National Natural Science Foundation of China: 62006009.

### Competing Interests

The authors declare there are no competing interests.

### Author Contributions

- Weiwei Feng conceived and designed the experiments, performed the experiments, analyzed the data, performed the computation work, prepared figures and/or tables, authored or reviewed drafts of the article, and approved the final draft.
- Yafang Li conceived and designed the experiments, analyzed the data, prepared figures and/or tables, authored or reviewed drafts of the article, and approved the final draft.
- Bo Li conceived and designed the experiments, authored or reviewed drafts of the article, and approved the final draft.
- Zhibin Jia conceived and designed the experiments, performed the experiments, analyzed the data, performed the computation work, prepared figures and/or tables, authored or reviewed drafts of the article, and approved the final draft.
- Zhihua Chu conceived and designed the experiments, authored or reviewed drafts of the article, and approved the final draft.

## Data Availability

The code is available at GitHub and Zenodo:

- https://github.com/vivian-166/BiMGCL

- vivian-166. (2023). vivian-166/BiMGCL: init (init). Zenodo. https://doi.org/10.5281/zenodo.7932312

The raw data is available in the Supplemental Files.

## Supplemental Information

Supplemental information for this article can be found online at http://dx.doi.org/10.7717/peerj-cs.1659#supplemental-information.

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
