# Peer review of "BiMGCL: rumor detection via bi-directional multi-level graph contrastive learning"

_PeerJ Computer Science, doi:10.7717/peerj-cs.1659_

## Round 0.1 · original submission · Major Revisions

After reading the reviewers' comments and the editor's reading, the paper still needs to be revised. Please revise the paper and resubmit it after careful revisions. Remember to highlight the revision parts with colours in the revised manuscript, together with point-by-point response letter.

The comments in detail provided by reviewers are given below.

The reviewers were unable to find the raw data. Please ensure that the raw data link is included in the main text of the manuscript, as well as in the data availability declaration.

Reviewer 1 ·

Basic reporting

no comment

Experimental design

no comment

Validity of the findings

no comment

Additional comments

Do more baseline comparisons.

Cite this review as

Reviewer 2 ·

Basic reporting

1. Writing is smooth.
2. BiMGCL used bi-directional graphs to represent the structural information of rumor propagation and proposed three graph strategy to augment the graphs. GIN was then used to learn the hidden representation of the graphs by exploiting a node-level and a graph-level contrastive loss.

Experimental design

The latest baseline RDEA was proposed in 2021, more recent baselines are expected to be compared

Validity of the findings

no comment

Additional comments

1. What is the intuition behind Edge Modification? Why would deleting and adding some edges help in addressing the rumor reappearance issue?
2. The lines between line 251 and equation (7) is inconsistent with Figure 1 and may cause confusion. ``for the rumor graph samples (A, X)``, this indicates that A contains all the adjacency matrices, and X contains all the feature matrices. However, only one rumor graph is presented in Figure 1. Besides, based on the description in the following paragraph after equation (7), it is natural to wonder why A is not A_i and X is not X_i?
3. What is the purpose of mentioning the dataset is ``few-shot`` in the description above equation (2)? Neither the datasets used in this paper were few-shot datasets, nor the few-shot idea was further mentioned in the subsequent sections.

Cite this review as

Reviewer 3 ·

Basic reporting

Literature references are outdated, irrelevant, and insufficient. Must be improved.
Raw data is not shared.

Experimental design

no comment

Validity of the findings

All the underlying data have not been provided
The conclusion section needs to be revised.

Additional comments

The manuscript submitted, “BiMGCL: Rumor detection via bi-directional multi level graph contrastive learning,” is interested in and has a broader scope in rumor detection from social media applications. The manuscript is well-written and well-organized, which shows the high quality of this manuscript, and I feel that this manuscript is fit for publication in PeerJ Computer Science. Here are my few concerns and suggestions to improve the quality of the manuscript.
• Last two rows in Table 1 needs more explanation for understating the posting and users.
• If datasets are publically available, provide online link to access it by readers.
• I recommend to add the implementation detail of the proposed framework. Also provide the link of the code to verify and regenerate the results. Further, to extend this work in future work.
• Line 312, provide a reference to justify the statement given in this line.
• Line 341, two or three observations?
• Figure 3, legends are hiding the graph lines, as a result difficult for readers to read and understand the results.
• In conclusion, add a short paragraph to include the future work and some limitations of the proposed framework.
• Read and cite the latest papers as noticed that only 5% references are from last 3 years.
• Did you perform three types of augmentation in parallel or sequential. Results in Table 2 and Table 3 doesn’t show the effects of augmentation and there is no discussion in the results section. I suggest the authors to perform augmentation analysis and add its results in the manuscript.
• Line 242, “To address the issue 3”, what is issue 3 and where are issue 1 and issue 2?
• This manuscript has a large number of mathematical equations that sounds good for understanding the methods but some equations needs more explanation like given below. I suggest to carefully revise all the equations.
o Small circles in equation 5 show which type of operation?
o Before equation 6, what are TD and BU in superscript?
o Equation 16, Equation 17 needs more explanation

Cite this review as

---

## Round 0.2 · accepted · Accept

The authors have addressed all the comments from the reviewers.

Reviewer 2 ·

Basic reporting

The revision has answered my questions

Experimental design

no comment

Validity of the findings

no comment

Cite this review as

Reviewer 3 ·

Basic reporting

All the concerns of the reviewers have been addressed. The second part of the second reviewer still needs to be answered by providing suitable references. "Why would deleting and adding some edges help in addressing the rumor reappearance issue".

There are still some language issues, like 51-line "Acquire"

Experimental design

improved after adding a few more experiments.

Validity of the findings

Findings are valid. Code and the datasets on GitHub show the validity of the findings mentioned in the manuscript.

Additional comments

No Comments

Cite this review as